# Segmentectomy Versus Wedge Resection for Stage IA Lung Adenocarcinoma—A Population-Based Study

**DOI:** 10.3390/cancers17060936

**Published:** 2025-03-10

**Authors:** Xu-Heng Chiang, Chih-Fu Wei, Ching-Chun Lin, Mong-Wei Lin, Chun-Ju Chiang, Wen-Chung Lee, Jin-Shing Chen, Pau-Chung Chen

**Affiliations:** 1Department of Medical Education, National Taiwan University Hospital, Taipei 100225, Taiwan; ntuhxhc@ntu.edu.tw; 2Department of Surgery, National Taiwan University Hospital and National Taiwan University College of Medicine, Taipei 100233, Taiwan; mwlin@ntu.edu.tw; 3Department of Environmental and Occupational Medicine, National Taiwan University Hospital Yun-Lin Branch, Yun-Lin 64041, Taiwan; y09486@ms1.ylh.gov.tw; 4Institute of Environmental and Occupational Health Sciences, College of Public Health, National Taiwan University Hospital, Taipei 100025, Taiwan; chingchun@ntu.edu.tw; 5Institute of Epidemiology and Preventive Medicine, College of Public Health, National Taiwan University, Taipei 106319, Taiwan; ruru.chiang@cph.ntu.edu.tw (C.-J.C.); wenchung@ntu.edu.tw (W.-C.L.); 6Taiwan Cancer Registry, Taipei 10055, Taiwan; 7Institute of Health Data Analytics and Statistics, College of Public Health, National Taiwan University, Taipei 106319, Taiwan; 8Department of Public Health, National Taiwan University College of Public Health, Taipei 10055, Taiwan; 9Department of Environmental and Occupational Medicine, National Taiwan University Hospital and National Taiwan University College of Medicine, Taipei 100233, Taiwan; 10National Institute of Environmental Health Sciences, National Health Research Institutes, Miaoli 35053, Taiwan

**Keywords:** non-small cell lung cancer (NSCLC), lung adenocarcinoma, sublobar resection, segmentectomy, wedge resection, population-based

## Abstract

This study utilizes Taiwan’s robust cancer registry to compare segmentectomy and wedge resection for Stage IA lung adenocarcinoma, the most common type of lung cancer. Results indicate that segmentectomy improves survival for tumors over 2 cm. Notably, this research addresses a critical gap in the knowledge left by two landmark trials, JCOG0802 and CALGB140503. While these trials established the non-inferiority of sublobar resection to lobectomy for small tumors, they did not definitively answer the question of whether segmentectomy or wedge resection is the superior sublobar approach. This study’s strength lies in its large, representative sample and rigorous propensity score matching, enabling reliable conclusions and informing individualized treatment strategies. By providing insights into the optimal surgical technique within the sublobar resection category, this research contributes valuable knowledge with the potential to improve patient care and inform future investigations.

## 1. Introduction

Lung cancer remains one of the leading causes of cancer-related mortality worldwide. Early diagnosis is crucial for improving survival rates, as it allows for timely intervention [1]. Lobectomy has been recommended as the standard treatment for early-stage lung cancer, particularly non-small cell lung cancer (NSCLC), due to its lower recurrence rate, as reported in the randomized controlled trial (RCT) comparing lobectomy and sublobar resection (SLR) for stage IA NSCLC by the Lung Cancer Study Group in 1995 [2,3].

In recent years, SLR, including segmentectomy and wedge resection (WR), has emerged as a viable alternative to lobectomy [4,5,6] due to its potential benefits, such as reduced postoperative complications and better preservation of pulmonary function [7]. Two recent large randomized controlled trials (RCTs), the Japanese Clinical Oncology Group trial (JCOG0802) and the Alliance trial (CALGB140503) in North America, have demonstrated that sublobar resection is not inferior to lobectomy for patients with stage IA NSCLC. However, neither study could determine the optimal method of SLR. The JCOG0802 trial only included anatomical segmentectomy (AS), while the Alliance trial lacked the power to compare outcomes of AS and WR [8,9].

Despite the growing interest in SLR, there is ongoing debate regarding its efficacy compared to lobectomy, especially in patients with stage IA lung NSCLC. The clinical outcomes of segmentectomy and WR remain equivocal, with some studies suggesting comparable survival rates, while others indicate a potential advantage for one technique over the other [10,11,12]. This underscores the need for further research to evaluate the differences in overall survival between these surgical approaches.

The increasing prevalence of computed tomography screening has led to a rise in the detection of smaller lung tumors [13]. This, coupled with advances in localization techniques [14], allows for adequate safety margins even with limited resection. Notably, the global surge in lung cancer diagnoses, particularly in East Asia, is predominantly driven by adenocarcinoma. Given that adenocarcinoma constitutes the majority of early-stage lung cancers requiring surgical resection, we focus on this subtype [15]. We hypothesize that for patients with stage IA lung adenocarcinoma, especially those with smaller tumors, wedge resection may be an appropriate alternative to segmentectomy as a sublobar resection.

This study aims to address the ongoing debate regarding the optimal sublobar resection technique for stage IA lung adenocarcinoma, specifically, segmentectomy versus wedge resection. By analyzing data from the Taiwan Cancer Registry Database (TCR), this research seeks to provide evidence-based guidance where current literature remains inconclusive. The findings of this study have the potential to influence clinical decision-making, particularly for patients who are unable to tolerate lobectomy. By providing evidence on the relative efficacy of WR and segmentectomy, this research aims to guide surgeons in selecting the most appropriate surgical approach for patients with early-stage lung cancer.

## 2. Materials and Methods

### 2.1. Study Population

This study focuses on patients diagnosed with stage IA lung adenocarcinoma between 2011 and 2018. Data from the Taiwan Cancer Registry (TCR) and the National Health Insurance Research Database (NHIRD) will be utilized for the analysis. To ensure a robust comparison between the two surgical techniques, the study will employ propensity score matching (PSM) and Cox regression models.

The study population was constructed from a secondary analysis of de-identified data from the TCR and NHIRD. The TCR, which has been collecting population-based data on newly diagnosed cancer cases from hospitals across Taiwan since 1979, is supported by the Ministry of Health and Welfare, Taiwan. Physicians in registered hospitals are required to report newly diagnosed cancer cases to the TCR, as well as follow-up conditions in the first and fifth years. Over the past decade, there have been no major changes in lung cancer reporting in the TCR, except for the inclusion of smoking status since 2011.

This study includes clinical stage IA adenocarcinoma cases diagnosed between 2011 and 2018, identified by ICD-O-3 codes: 8140–8149, 8250–8269, 8450–8459, 8480–8489, 8520–8529, 8550–8559, and 8570–8579. Excluded from the study were cases of squamous cell carcinoma (8050–8059, 8070–8079, 8080–8089), adenosquamous carcinoma (8560–8569), large cell carcinoma (8000–8009, 8010–8019, 8310–8319), and carcinoid tumor (8240–8249). Patients with other primary cancers, incomplete information on surgical modality, positive surgical margins, clinically metastatic ipsilateral or contralateral mediastinal, supraclavicular/scalene lymph nodes, and distant metastasis were also excluded.

Additionally, information on inpatient care, intensive care unit use, and prescribed treatments was obtained from the NHIRD. The NHIRD contains data on administrative and health claims from Taiwan’s national health insurance program. Individuals were linked using scrambled identification numbers associated with their relevant claims information. The selection algorithm is demonstrated in Figure 1.

### 2.2. Surgical Modalities and Main Outcomes

The main exposure status was the surgical modalities for lung cancer, categorized as segmentectomy or WR. The operation type for each patient was reviewed by two physicians, and a third physician determined the final type in cases of disagreement. Operations without free surgical margins were excluded to ensure that incomplete resection did not interfere with the outcomes.

The main outcome was the overall survival after surgery for lung cancer, defined as the interval between surgery and mortality, as patients in the TCR were actively followed by the reported hospitals in the first and fifth years after initial reporting. We further linked mortality data, including survival status and mortality date, from the TCR to the NHIRD. This allowed us to obtain survival dates, as the NHIRD includes mortality information by linking with the death registry of Taiwan.

### 2.3. Confounders and Covariates

Possible confounding factors were selected using scientific approaches based on reviewing the previous literature on survival in early-stage adenocarcinoma. The initial set of variables was decided after a panel discussion including senior epidemiologists and chest surgeons with a modified Delphi method. Meanwhile, a direct acyclic graph was employed to decide the parsimonious set of confounders in multivariable models using DAGitty v3.0. Smoking history (never, quit, or current smoker), age at operation (above or below 75 years), sex (male or female), differentiation status of the tumor (well differentiated, moderately differentiated, poor or undifferentiated, or unknown), pathological stage (including pathological T and N staging), and tumor size (less than 1 cm, 1–2 cm, 2–3 cm, and above 3 cm) were included as confounders in our model.

### 2.4. Propensity Score Matching

We conducted PSM in this study. The same baseline covariates were used in both PSM models, selected according to the previous literature [16]. Matching variables included age (above or below 75 years and the chronological age), tumor size (both in continuous and categorical scales), pathological stage (both in primary tumor, pT and lymph node involvement, and pN), lymphovascular invasion (LVI), visceropleural invasion (VPI), sex (male or female), histological diagnosis and subtypes (acinar, papillary, micropapillary, and others), and self-reported smoking history (never, quit, or current smoker). Exact matching was applied to tumor size categories and age groups because they were found to be highly correlated with patient survival and surgery choices. Baseline covariates were matched at a 1:1 ratio using a greedy matching algorithm, and the caliper was set at 0.01. The goodness-of-fit of the model was assessed using the Hosmer–Lemeshow test and c-statistics, while the distribution of the logit propensity scores was examined with a cloud plot (Appendix A).

### 2.5. Descriptive Statistics

The same approach was employed for the entire study population and the PS-matched subpopulation for descriptive statistical analyses. Continuous variables are presented as means and standard deviations (SD). Meanwhile, categorical variables in each treatment group are presented as numbers and percentages. Anova (one-way) was used to identify differences in various outcomes between the two methods. The overall survival of patients undergoing different types of operations was evaluated using Kaplan–Meier curves, with *p*-values estimated using the log-rank test. A *p* value < 0.05 represented statistical significance.

### 2.6. Survival Analysis and Sensitivity Analysis

The Cox proportional hazards model with a stratified baseline hazard across different tumor size categories was adapted to estimate crude and adjusted hazard ratios (aHRs) for overall survival after surgical intervention. HRs were reported to compare segmentectomy and WR in the entire study population.

Subgroup analyses were conducted to examine effect heterogeneity within the study population based on tumor size. Baseline hazard stratification was employed for these variables, considering that the proportional hazard assumption may not hold across their levels.

## 3. Results

### 3.1. Baseline Characteristics

In this retrospective study, data on 6598 patients with clinical stage IA lung adenocarcinoma undergoing SLR between 2011 and 2018 were collected from the TCR, including 2061 and 4537 cases of segmentectomy and WR, respectively. The mean age and body mass index (BMI) of this cohort were 60.3 ± 11.7 years and 24.1 ± 3.7 kg/m^2^, respectively, with 66.2% being female. Patients who never smoked accounted for 81.5% of this cohort. The tumor size was less than 1 cm, between 1 and 2 cm, and larger than 2 cm in 47.2%, 39.7%, and 13.1% of the patients, respectively. Approximately half (48.0%) of the lung adenocarcinomas were lepidic-predominant. Among patients with lung adenocarcinoma, 0.8% and 8.3% showed LVI and pleural invasion, respectively. Approximately 1.3% of patients with clinical stage IA lung adenocarcinoma were found to have nodal metastasis.

Before PSM, patients undergoing WR had less advanced and less invasive cancer features than those undergoing segmentectomy, including smaller tumor size (1.1 ± 0.7 cm vs. 1.3 ± 0.8 cm, *p* < 0.001), lower rates of nodal metastasis (1% vs. 2%, *p* < 0.001), better cell differentiation (well-differentiated: 42.8% vs. 39.6%, *p* < 0.001), and a higher proportion of the lepidic-predominant histological subtype (50.3% vs. 43.1%, *p* < 0.001). In addition, the number of examined nodes was less in the WR group than in the segmentectomy group (6.3 ± 7.0 vs. 12.8 ± 9.2, *p* < 0.001).

The baseline demographic data of the unmatched and two PS-matched populations are shown in Table 1.

### 3.2. Overall Survival

The overall survival analysis showed that segmentectomy was associated with better survival in the entire cohort (*p* < 0.001, Figure 2A). In patients with a tumor less than 1 cm, no difference in survival was observed between segmentectomy and WR (*p* = 0.051, Figure 2B). In patients with a tumor between 1 and 2 cm, segmentectomy demonstrated a significant survival advantage over WR (*p* < 0.001, Figure 2C). In patients with a tumor larger than 2 cm, the results aligned with those in the entire cohort, as segmentectomy was associated with significantly greater survival. (*p* < 0.001, Figure 2D).

### 3.3. Propensity Score Matching for Survival Analysis

To balance baseline characteristics (e.g., age, tumor severity, etc.) between the segmentectomy and WR groups, 1499 pairs were selected from 6598 patients using PSM (Appendix A). After PSM, no significant differences in baseline characteristics were observed between the segmentectomy and WR groups (Table 1).

Kaplan–Meier analysis showed superior overall survival in the segmentectomy group compared to that in the WR group (*p* = 0.019, Figure 3A). However, in patients with a tumor smaller than 2 cm, similar survival outcomes were noted (*p* = 0.31 for tumors 0–1 cm and *p* = 0.38 for tumors 1–2 cm, Figure 3B,C). In patients with a tumor larger than 2 cm, Kaplan–Meier plots were consistent with the results of the entire post-matching cohort (*p* = 0.038, Figure 3D), as segmentectomy was characterized by significantly greater survival in patients with a tumor 2 cm or larger. A subgroup analysis of patients with tumors ≤ 2 cm also demonstrated comparable overall survival between segmentectomy and wedge resection (Appendix A). The lung cancer-specific survival analysis (Appendix A) yielded results that were consistent with the overall survival analysis, revealing a parallel trend.

### 3.4. Cox Proportional Hazards Models for Overall Survival

In the univariate Cox regression analysis, all of the included factors were significant predictors of overall survival, including surgical methods, smoking status, age, sex, tumor differentiation, tumor size, pathological N staging, visceral pleural invasion, and LVI. On the other hand, multivariate analysis showed that surgical methods, smoking status, age, tumor differentiation, tumor size, and pathological N2 stage were significant risk factors for poor overall survival (Table 2).

## 4. Discussion

In this population-based study, we found that segmentectomy was significantly associated with better overall survival than WR for patients with lung adenocarcinoma with tumors 2 cm or larger. For lung adenocarcinomas smaller than 2 cm, survival outcomes were comparable between WR and segmentectomy. Aged 75 years or younger, well-differentiated tumors, smaller tumor size, and absence of nodal metastasis were also significantly associated with better overall survival.

Our findings are consistent with the survival trends reported by Cao [17], who suggested that the outcomes of lobectomy, segmentectomy, and WR were comparable for tumors ≤ 1.0 cm in stage IA NSCLC. For tumors 1–2 cm, lobectomy and segmentectomy showed similar survival rates, both superior to WR. For tumors 2–3 cm, lobectomy remained the standard treatment; however, if lobectomy was unsuitable, segmentectomy and WR showed similar outcomes [17]. These findings are supported by a previous single-institutional study conducted by our team [16]. As expected, less extensive resection achieves comparable survival outcomes for smaller sub-centimeter lung nodules [18]. The analysis of the Society of Thoracic Surgeons General Thoracic Surgery Database suggested similar effectiveness between WR and segmentectomy for early lung NSCLC [19]. Collectively, these studies indicate a trend: in early-stage non-small cell lung cancer, the smaller the tumor, the less significant the differences in oncological outcomes between surgical techniques. However, as tumor size increases, anatomical resection becomes more crucial to ensure adequate resection margins.

Nevertheless, data from two prior studies refute these results, as they exhibit a significant association between anatomical resection and higher long-term survival [6,20]. However, size stratification analysis was not performed in either study. Recent advancements in lung cancer staging have led to more detailed classifications. For example, traditional stage IA encompasses a wide range of patients with significantly different prognoses. Therefore, stratification analyses based on tumor size are essential for accurately addressing the issue of surgical resection choices for lung cancer.

Two recent RCTs showed that SLR, including segmentectomy and WR, was not inferior to lobectomy in promoting survival for stage IA lung cancer measuring 2 cm or smaller [8,9]. With the increased use of computed tomography (CT) screening, SLR may become the standard treatment for these patients [21]. Although a re-analysis of the CALBG 140503 trial showed no significant difference in survival probability between WR and segmentectomy, post hoc analysis has limitations in providing definitive conclusions on this issue [22]. Post hoc analysis, while valuable, cannot definitively answer questions that were not designed to address the original RCT, as the trial is conducted after data collection and often involves exploring patterns that were not predefined. This can lead to issues such as multiple comparisons and an increased risk of Type I errors, where results may appear significant by chance rather than reflecting a true effect [23]. Additionally, post hoc analysis may lack sufficient statistical power and potential biases may be introduced by the retrospective nature of the analysis [24].

To address the questions that the aforementioned RCTs have yet to answer—specifically, while it is established that sublobar resection is an appropriate surgical option for stage IA NSCLC, it remains unclear whether segmentectomy or WR is the preferable choice—we conducted this population-based study. Additionally, this study aims to further validate the findings from our previous single-center research, which indicated that for tumors larger than 2 cm, segmentectomy provides better outcomes than WR.

Analyzing a larger and more diverse patient population with an appropriate study design allowed us to validate findings from RCTs in a real-world setting and provides insights that are generalizable to a broader patient population. We utilized the TCR and NHIRD, which are maintained and regulated rigorously by the governmental unit, the Data Science Centre of the Ministry of Health and Welfare of Taiwan [25]. Furthermore, PSM was employed to effectively balance differences in baseline characteristics between the two groups, enhancing the reliability of our results. Additionally, we considered various potential confounding factors in our model, which strengthens the validity of our conclusions.

While most studies on early-stage lung cancer surgery have focused on NSCLC as a whole, this study specifically examined patients diagnosed with adenocarcinoma. This choice was based on several factors. First, squamous cell carcinoma (SCC) and adenocarcinoma, the two main subtypes of NSCLC, exhibit significant differences in clinical presentation and behavior. For example, Wang et al. (2022) found that patients with SCC had a higher proportion of males, smokers, and those with higher-stage tumors compared to patients with adenocarcinoma [26]. Second, adenocarcinoma is the most common histotype for early-stage lung cancer in East Asia, occurring three to four times more frequently than SCC [27]. This makes adenocarcinoma a particularly relevant subtype to study in this region. By focusing on adenocarcinoma, this study aimed to analyze a more homogenous patient population. However, the study’s findings suggest that the choice of surgical approach for early-stage lung adenocarcinoma aligns with the broader recommendations for NSCLC. Specifically, for tumors less than 2 cm, segmentectomy and wedge resection resulted in comparable survival outcomes.

While this study acknowledges the potential for lower lymph node counts in wedge resections compared to segmentectomy or lobectomy, this difference does not appear to negatively affect survival outcomes in patients with early-stage lung cancer, especially with propensity score matching to minimize bias. This aligns with the literature suggesting that extensive lymph node dissection may not confer a survival benefit in this population. Furthermore, a less aggressive approach to lymph node dissection could potentially reduce the risk of complications, as supported by findings from the ACOSOG Z0030 trial [28]. Therefore, in the context of early-stage lung cancer, the extent of lymph node dissection may not be the sole determinant of survival outcomes and might not be the most critical factor when selecting the surgical approach.

While our findings regarding comparable survival between segmentectomy and atypical resection for ≤2 cm tumors are consistent with the prior literature, our study uniquely reinforces these observations within a large East Asian adenocarcinoma cohort, the predominant lung cancer subtype in this region and globally [29,30]. Utilizing rigorous propensity score matching on a large-scale database, we provide robust, real-world validation of existing conclusions. This targeted approach strengthens the clinical relevance and generalizability of our results, offering clinicians a reliable reference for surgical decision-making in this specific patient population.

Despite the comprehensive and detailed information provided by the TCR, there are several limitations to our study. One significant limitation is the absence of imaging information, such as the consolidation-to-tumor ratio, which is crucial for predicting the prognosis of lung cancer [21]. Moreover, the TCR database does not provide tumor depth data, a factor that may influence surgeons’ choices regarding surgical procedures. Additionally, the database does not include information on resection margins, making it difficult to assess the quality of SLRs. Furthermore, the database does not include important perioperative details such as operation time, postoperative complications, and length of hospital stay. Lastly, our study lacks granular data on the cause of death, which could introduce confounding factors related to comorbidities when analyzing survival outcomes. This limitation stems from the inherent constraints of using secondary data sources like the TCR and NHIRD, which may not capture the full spectrum of patient-specific factors influencing mortality. The lack of information hinders verification of whether WR offers better perioperative outcomes than segmentectomy, as we expected. Moreover, the database does not provide data on preoperative assessments, such as pulmonary function, which are essential for evaluating surgical candidacy, making it challenging to identify potential unknown biases in surgical choices within this cohort. These limitations highlight the need for more comprehensive data collection to fully understand and compare the outcomes of different surgical approaches for emerging lung adenocarcinoma.

## 5. Conclusions

In summary, this study provides evidence supporting the advantages of segmentectomy in specific patient populations. Our findings not only aid clinical decision-making but also lay a foundation for future research. For lung adenocarcinomas larger than 2 cm, we recommend segmentectomy due to its association with better overall survival than WR. Additionally, for tumors smaller than 2 cm, our results indicate that segmentectomy and WR offer comparable outcomes. This nuanced understanding can help tailor surgical approaches to individual patient needs, ultimately enhancing patient care and outcomes.

## Figures and Tables

**Figure 1 cancers-17-00936-f001:**
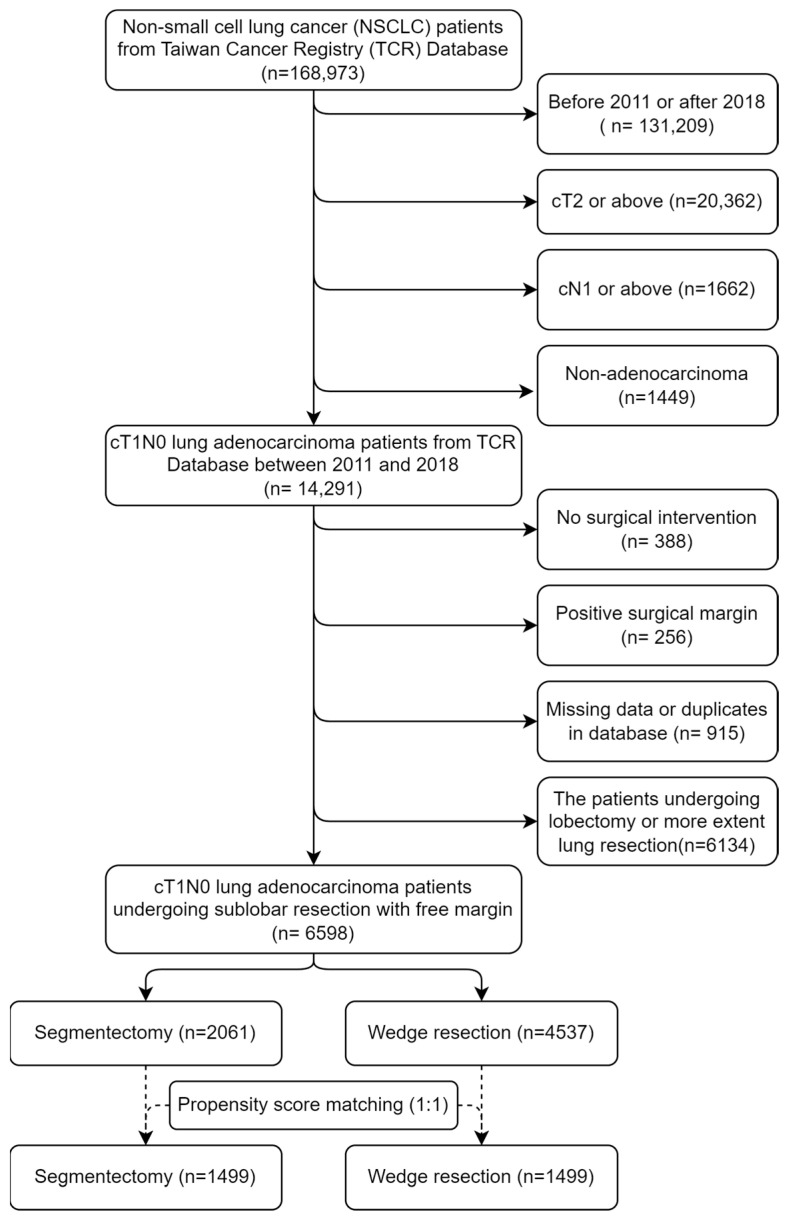
Selection algorithm.

**Figure 2 cancers-17-00936-f002:**
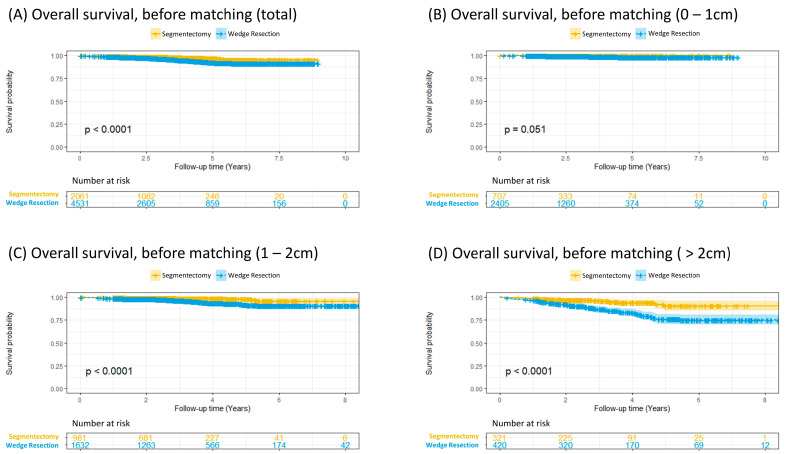
Overall survival analysis before matching; (**A**) for total included patients; (**B**) for tumor size 0–1 cm; (**C**) for tumor size 1–2 cm; (**D**) for tumor size > 2 cm.

**Figure 3 cancers-17-00936-f003:**
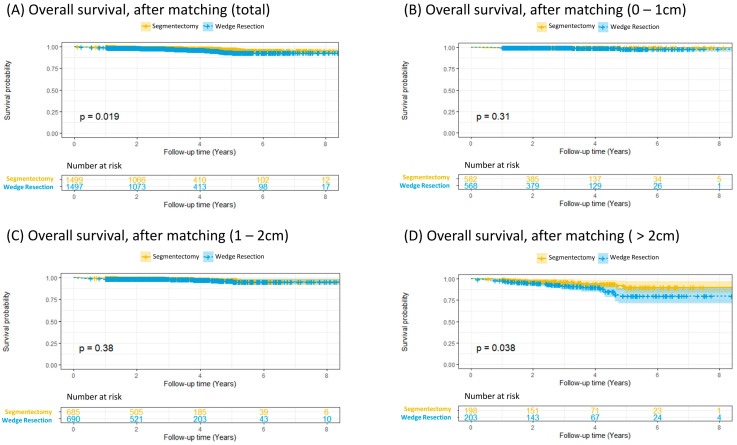
Overall survival analysis after matching; (**A**) for total included patients; (**B**) for tumor size 0–1 cm; (**C**) for tumor size 1–2 cm; (**D**) for tumor size > 2 cm.

**Table 1 cancers-17-00936-t001:** Demographic and clinical features of clinical stage IA lung adenocarcinoma patients undergoing sublobar lung resection.

	Before Matching				After Matching			
	All	Segmentectomy	Wedge	*p* Value	All	Segmentectomy	Wedge	*p* Value
	n = 6598	n = 2061	n = 4537		n = 2998	n = 1499	n = 1499	
Age, yr	60.3 (11.7)	60.9 (11.0)	60.1 (12.0)	0.012	61.01 (11.54)	60.77 (11.25)	61.26 (11.83)	0.248
Female	4371 (66.2)	1348 (65.4)	3023 (66.6)	0.344	1957 (65.3)	987 (65.8)	970 (64.7)	0.539
BMI	24.1 (3.7)	24.2 (3.5)	24.0 (3.8)	0.136	24.07 (3.58)	24.12 (3.50)	24.01 (3.66)	0.403
Smoking status				0.412				0.739
Never smoked	5375 (81.5)	1676 (81.3)	3699 (81.5)		2462 (82.1)	1227 (81.9)	1235 (82.4)	
Smoked	1203 (18.2)	376 (18.2)	827 (18.2)		536 (17.9)	272 (18.1)	264 (17.6)	
Tmor size (cm)				<0.001				0.955
0–1	3114 (47.2)	707 (34.3)	2407 (53.1)		1150 (38.4)	582 (38.8)	568 (37.9)	
1–2	2617 (39.7)	981 (47.6)	1636 (36.1)		1377 (45.9)	685 (45.7)	692 (46.2)	
2–3	741 (11.2)	321 (15.6)	420 (9.3)		401 (13.4)	198 (13.2)	203 (13.5)	
≥3	126 (1.9)	52 (2.5)	74 (1.6)		70 (2.3)	34 (2.3)	36 (2.4)	
Differentiation *				<0.001				0.814
Well	2757 (41.8)	816 (39.6)	1941 (42.8)		1220 (40.7)	617 (41.2)	603 (40.2)	
Moderate	2919 (44.2)	1023 (49.6)	1896 (41.8)		1452 (48.4)	716 (47.8)	736 (49.1)	
Poor	296 (4.5)	96 (4.7)	200 (4.4)		145 (4.8)	71 (4.7)	74 (4.9)	
Histology subtype				<0.001				0.660
Lepidic	3170 (48.0)	888 (43.1)	2282 (50.3)		1351 (45.1)	669 (44.6)	682 (45.5)	
Acinar	1797 (27.2)	697 (33.8)	1100 (24.2)		939 (31.3)	487 (32.5)	452 (30.2)	
Papillary	304 (4.6)	102 (4.9)	202 (4.5)		157 (5.2)	74 (4.9)	83 (5.5)	
Micropapillary	67 (1.0)	28 (1.4)	39 (0.9)		39 (1.3)	21 (1.4)	18 (1.2)	
Solid	1260 (19.1)	346 (16.8)	914 (20.1)		512 (17.1)	248 (16.5)	264 (17.6)	
Lymphovascular invasion *	54 (0.8)	25 (1.2)	29 (0.6)	0.005	29 (1.0)	13 (0.9)	16 (1.1)	0.709
Visceral pleural invasion *				0.007				0.931
PL0	6049 (91.7)	1874 (90.9)	4175 (92.0)		2729 (91.0)	1368 (91.3)	1361 (90.8)	
PL1	364 (5.5)	140 (6.8)	224 (4.9)		189 (6.3)	93 (6.2)	96 (6.4)	
PL2	125 (1.9)	33 (1.6)	92 (2.0)		55 (1.8)	27 (1.8)	28 (1.9)	
PL3	60 (0.9)	14 (0.7)	46 (1.0)		25 (0.8)	11 (0.7)	14 (0.9)	
pT stage				<0.001				0.838
T1a	3354 (50.8)	789 (38.3)	2565 (56.5)		1242 (41.4)	633 (42.2)	609 (40.6)	
T1b	2224 (33.7)	853 (41.4)	1371 (30.2)		1207 (40.3)	597 (39.8)	610 (40.7)	
T1c	447 (6.8)	217 (10.5)	230 (5.1)		263 (8.8)	128 (8.5)	135 (9.0)	
T2 and above	571 (8.7)	202 (9.8)	369 (8.1)		286 (9.5)	141 (9.4)	145 (9.7)	
pN stage *				<0.001				0.612
N0	5557 (84.2)	1971 (95.6)	3586 (79.0)		2862 (95.5)	1435 (95.7)	1427 (95.2)	
N1	20 (0.3)	14 (0.7)	6 (0.1)		8 (0.3)	5 (0.3)	3 (0.2)	
N2	67 (1.0)	26 (1.3)	41 (0.9)		32 (1.1)	13 (0.9)	19 (1.3)	
Nx	954 (14.5)	50 (2.4)	904 (19.9)		96 (3.2)	46 (3.1)	50 (3.3)	
Adjuvant therapy				0.002				0.770
CCRT	18 (0.3)	0 (0.0)	18 (0.4)		0			
Chemotherapy	286 (4.3)	99 (4.8)	187 (4.1)		148 (4.9)	73 (4.9)	75 (5.0)	
Radiotherapy	23 (0.3)	2 (0.1)	21 (0.5)		4 (0.1)	2 (0.1)	2 (0.1)	
Target therapy	73 (1.1)	18 (0.9)	55 (1.2)		23 (0.8)	9 (0.6)	14 (0.9)	
Lymphnode dissection								
Nodes examined	8.30 (8.32)	12.78 (9.15)	6.27 (7.03)	<0.001	10.55 (7.60)	10.47 (7.48)	10.62 (7.72)	0.599

Data are presented as mean ± SD (range) or number (%). * The categories of differentiation, lymphovascular invasion, visceral pleural invasion, and pathological stage are lacking some data. CCRT, concurrent chemoradiotherapy; ICU, intensive care unit; LN, lymph node.

**Table 2 cancers-17-00936-t002:** Cox regression analyses of correlations between clinicopathological features and overall survival for the clinical stage IA lung adenocarcinoma patients undergoing sublobar lung resection.

Variables	Overall Mortality					
	Univariate Analysis			Multivariate Analysis		
	Hazard Ratio	95% CI	*p* Value	Hazard Ratio	95% CI	*p* Value
Surgical method						
Segmentectomy	1			1		
Wedge resection	2.17	1.51–3.12	<0.001	2.16	1.44–3.24	<0.001
Smoking						
Never smoked	1			1		
Smoked	3.45	2.64–4.50	<0.001	1.92	1.25–2.97	0.003
Age						
≤75 years-old	1			1		
>75 years-old	3.69	2.80–4.86	<0.001	2.10	1.46–3.03	<0.001
Gender						
Male	1			1		
Female	0.41	0.31–0.53	<0.001	0.69	0.45–1.06	0.089
Differentiation						
Well	1			1		
Moderate	2.59	1.84–3.64	<0.001	1.88	1.15–3.05	0.011
Poor	10.08	6.68–15.20	<0.001	3.11	1.70–5.70	<0.001
Size						
0–1 cm	1			1		
1–2 cm	4.31	2.73–6.81	<0.001	3.90	1.89–7.64	<0.001
2–3 cm	13.34	8.41–21.15	<0.001	8.23	3.96–17.12	<0.001
>3 cm	20.60	11.62–36.54	<0.001	10.61	4.48–25.16	<0.001
Pathological N stage						
N0	1			1		
N1	4.63	1.14–18.72	0.032	1.74	0.42–7.27	0.448
N2	14.63	8.80–24.34	<0.001	4.91	2.73–8.84	<0.001
Visceral pleural invasion						
No	1			1		
Yes	4.09	3.07–5.46	<0.001	1.40	0.94–2.08	0.099
Lymphovascular invasion						
No	1			1		
Yes	4.55	1.44–14.38	0.010	0.66	0.09–4.95	0.690

CI, confidence interval.

## Data Availability

Restrictions apply to the availability of these data. Data were obtained from Taiwan Cancer Registry Database and are available with the permission of Taiwan Ministry of Health and Welfare.

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
