# Peer review of "Segmentectomy Versus Wedge Resection for Stage IA Lung Adenocarcinoma—A Population-Based Study"

_cancers, 2025, doi:10.3390/cancers17060936_

Round 1

Reviewer 1 Report

Comments and Suggestions for Authors

In the article 'Segmentectomy versus Wedge Resection for Stage IA Lung Adenocarcinoma - A Population-based Study', the authors describe an analysis of the results of segmentectomy and wedge resection in patients with stage cIA lung adenocarcinoma from the Taiwan Cancer Registry. Previous studies have shown that although lobectomy is the gold standard treatment, segmentectomy provides oncological outcomes comparable to lobectomy for tumours smaller than 2 cm. Alternatively, segmentectomy is indicated for patients with compromised lung function or significant comorbidities (Bottet, B.; Hugen, N.; Sarsam, M.; Couralet, M.; Aguir, S.; Baste, J.-M. Performing High-Quality Sublobar Resections: Key Differences Between Wedge Resection and Segmentectomy. Cancers 202416, 3981. https://doi.org/10.3390/cancers16233981).

I have a few suggestions:

- The authors report: "While these trials demonstrated the non-inferiority of sublobar resection to lobectomy for small tumours, they did not definitively answer the question of whether segmentectomy or wedge resection is the superior sublobar approach. However, in their comparative analysis, the authors included patients with cIa adenocarcinoma, including tumours with a diameter of 2-3 cm, for which previous studies have shown that lobectomy is associated with better oncological outcomes. Therefore, the authors should focus their analysis on tumours < 2 cm or alternatively separate the two groups (tumours < 2 cm and tumours 2-3 cm), as in the second group sublobar resection should only be offered to high risk patients.

- In the discussion, the authors should highlight the overlapping results obtained between wedge resections and segmentectomies in tumours < 1 cm and compare these results with those obtained in other studies.

- The patients selected for this study were all diagnosed with adenocarcinoma, making the sample more homogeneous. Were the results different from other trials that included patients with other histotypes?

Reviewer 2 Report

Comments and Suggestions for Authors

I would like to commend Drs Chiang, Wei, et al., on their well designed and completed study.  The main objective of this study was to compare the impact of segmentectomy versus edge resection in Stage I lung adenocarcinoma.  It has been well established over the past few years, with three randomized controlled trials that sublobar resections are equivalent to lobectomy for early stage peripheral lung cancers, however the benefit or lack thereof for segmentectomy versus wedge resection has been elusive. This paper evaluated 6598 patients, through the Taiwanese Cancer Registry, between 2011 and 2018, who underwent a segmentectomy or wedge resection for T1N0 lung adenocarcinoma, and then performed propensity score matching to generate 1499 matched pairs. They concluded that there was an improvement in lung cancer specific survival in those patients who underwent a segmentectomy if their tumor was greater than 2cm in size. 

This study represents a retrospective review of the treatment of lung cancer in the national cancer registry in Taiwan.  The author’s findings of similar disease specific mortality between the two groups for tumors smaller than 2cm is widely supported by the recent randomized control trials. The conclusions of this paper, mirror those in other retrospective studies. The data presented in this study support the overall conclusions, and I surmise that a more discrete cutoff for the benefit of segmentectomy versus wedge resection will be generated as further studies are performed in this area.  This study helps to lay some of the framework for that discussion.

There are several potentially significant biases to a retrospective study, and the authors comment on this fact. The lack of consolidation tumor ratio is particularly challenging, because with purely ground glass or mostly ground glass lesions the risk for recurrence is so low, that it would be unlikely that the surgical treatment has much impact. Furthermore, there is potential for significant selection bias as to the location of the tumor within the lung such that wedge resections were performed for more peripheral tumors while segmentectomies were performed for more central tumors and have inherently higher risk profiles for recurrence.

I would be interested in some more information about the overall survival between the two cohorts and if there was any difference there. One of the primary takeaways from the recent randomized control trials is that the lobectomy patients seem to have a higher rate of non cancer related mortality compared to the segmentectomy group.  This study focusing exclusively on cancer related mortality may miss out on an outcome such as that.

I would also draw attention to the difference in lymph node dissection pre propensity matching.  This is a drawback of telling people that wedge and segmentectomy are equivalent, because most people do not do an equivalent lymph node dissection, and in your group the segments got twice as many lymph nodes examined on average compared to the wedge group.

Overall, I enjoyed reading this well written article, and I would like to thank the authors for their manuscript and contribution to the treatment of this challenging disease process.

Reviewer 3 Report

Comments and Suggestions for Authors

Dear authors,

The results of this retrospective analysis of 6598 patients' data across two well defined and organized national databases (TCR and NHIRD) revealed that after propensity score matching (PSM), segmentectomy for lung adenocarcinoma  at clinical stage Ia was associated with better survival compared to wedge resection (WR) in tumors larger than 2 cm. The survival between the two groups was similar when dealing with adenocarcinomas smaller than 2cm.

The strength of a study is its large sample, enabling propensity score matching to balance baseline differences. Congratulations for your extensive work.

However, the weakness is its retrospective nature. Very important, the reasons behind  the surgeons’ choice of specific type of operation remain unclear. It is presumed, that patients selected for WR may have been in poorer psychophysical condition and had more comorbidities, factors that could have significantly affected their survival.

Major comment:

-          The authors chose lung cancer specific survival as the primary endpoint, but mortality date is the only data mentioned about survival. Do you have information on the cause of death, or the time of potential lung cancer progression? If not, the observed differences in survival could be influenced by comorbidities rather than cancer progression, and this should be addressed in the discussion.

-          The reported patients’ characteristics are limited to age, smoking status, sex, BMI and tumor characteristics. It would be helpful to include more detailed information on important comorbidities, lung function, WHO PS.

Reviewer 4 Report

Comments and Suggestions for Authors

I would like to commend the authors for their effort in conducting this retrospective analysis. In the treatment of non-small cell lung cancer, a minimally invasive approach and lung tissue preservation are, to date, the primary endpoints.

Xu-Heng Chiang et al., in their study, conclude that for tumors measuring ≤1 cm, there is no significant difference in terms of survival between segmentectomy and atypical resection. However, these conclusions are already widely supported by the literature, particularly for lepidic and well-differentiated tumors, which account for almost 50% of the population analyzed in this study.

Moreover, the study shows some gaps in data collection. Specifically, the authors refer to atypical resections without providing details on the amount of lung parenchyma removed, its proportion relative to the tumor size, or the distance of the surgical margins from the tumor.

Considering the limited originality of the conclusions and the methodological shortcomings highlighted, I believe that the study, in its current form, cannot be accepted.

Comments on the Quality of English Language

The English could be improved to more clearly express the research.

Round 2

Reviewer 4 Report

Comments and Suggestions for Authors

The study presents a significant limitation in the data collected and lacks originality. It is certainly a study with a rich sample, but unfortunately, the data is not optimized. I believe that the article cannot be accepted in its current state.